# DisPose: Disentangling Pose Guidance for Controllable Human Image Animation

**Hongxiang Li**[1], **Yaowei Li**[1], **Yuhang Yang**[2], **Junjie Cao**[3], **Zhihong Zhu**[1],
**Xuxin Cheng**[1], **Long Chen**[4][†]

[1]Peking University    [2]University of Science and Technology of China
[3]Tsinghua University    [4] Hong Kong University of Science and Technology

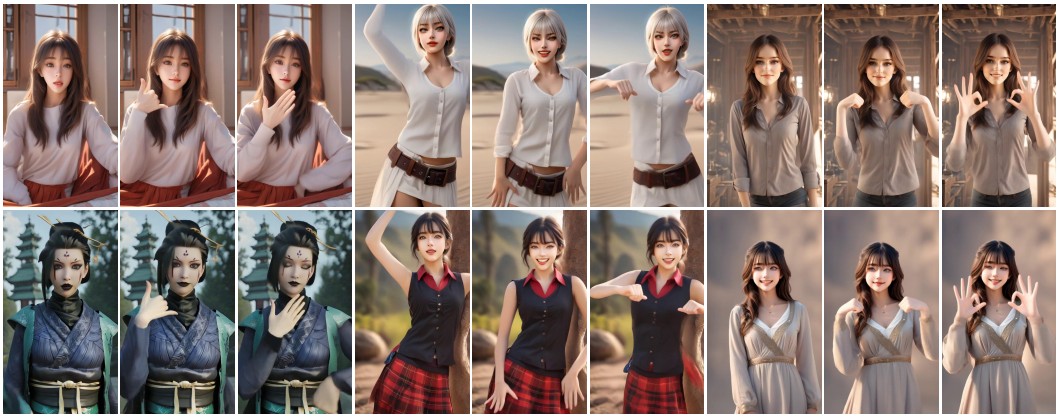

Figure 1: Our method demonstrates its ability to produce diverse animations and preserve consistency of appearance.

## Abstract

Controllable human image animation aims to generate videos from reference images using driving videos. Due to the limited control signals provided by sparse guidance (e.g., skeleton pose), recent works have attempted to introduce additional dense conditions (e.g., depth map) to ensure motion alignment. However, such strict dense guidance impairs the quality of the generated video when the body shape of the reference character differs significantly from that of the driving video. In this paper, we present DisPose to mine more generalizable and effective control signals without additional dense input, which disentangles the sparse skeleton pose in human image animation into motion field guidance and keypoint correspondence. Specifically, we generate a dense motion field from a sparse motion field and the reference image, which provides region-level dense guidance while maintaining the generalization of the sparse pose control. We also extract diffusion features corresponding to pose keypoints from the reference image, and then these point features are transferred to the target pose to provide distinct identity information. To seamlessly integrate into existing models, we propose a plug-and-play hybrid ControlNet that improves the quality and consistency of generated videos while freezing the existing model parameters. Extensive qualitative and quantitative experiments demonstrate the superiority of DisPose compared to current methods. Project page: https://github.com/lihxxx/DisPose.

## 1 Introduction

Controllable video generation (Zhang et al., 2023b; Yin et al., 2023; Wang et al., 2024c) has gained increasing attention for its ability to customize videos based on user preferences. In particular,

---

[†] Corresponding author.

controllable human image animation (Hu et al., 2023; Wang et al., 2024b) has attracted significant interest due to its vast potential applications in art creation, social media, and digital humans Ma et al. (2018); Xu et al. (2024b); Ma et al. (2017); Yang et al. (2024b); Xu et al. (2024c). It aims to animate static images into realistic videos based on driving videos. In contrast to other controllable video generation methods (e.g., camera control (He et al., 2024; Wang et al., 2024c), motion control (Wu et al., 2024; Wang et al., 2024c; Li et al., 2024; Jiang et al., 2024; Niu et al., 2024)), controllable human image animation allows for more flexible motion regions, diverse motion paradigms, and complex character appearances. Introducing precise pose control in existing video generation methods is challenging, but has significant value in achieving the desired results. There are two challenges: (1) following the motion of the driving video, and (2) preserving the appearance information of the reference image.

The motion control signal is critical to drive the animation. Controllable human image animation usually utilizes the skeleton pose (Yang et al., 2023) as the control signal. Besides the fact that the skeleton pose is easy to obtain, the more important reason is that it is easier to adapt to different body shapes when the target body shape is significantly different from the reference image. However, the skeleton pose as a sparse expression provides limited guidance information. To provide more control signals, recent work (Zhu et al., 2024; Xu et al., 2024a) has attempted to express human body geometry and motion regions by extracting various dense signals from the driving video, including DensePose (Güler et al., 2018), 3D human motion (Loper et al., 2015) and depth map (Yang et al., 2024a), etc. Unfortunately, these dense signals impose strict shape constraints on the generated characters, they are more difficult to adapt to reference images with different body shapes (Jin et al., 2024). Moreover, extracting dense signals accurately in complex motion videos is inherently difficult (Zhang et al., 2024). These overly dense guidance techniques exacerbate pose estimation errors and thus impair generation quality. Therefore, the existing methods (Chang et al., 2023; Hu et al., 2023; Zhu et al., 2024; Wang et al., 2024b) are struggling to trade off generalizability and effectiveness between sparse and dense controls. It would be beneficial to mine more generalizable and effective control signals from the skeleton pose map instead of dense control inputs.

On the other side, preserving appearance consistency from complex motion is also extremely challenging. Image-driven generation methods (Zhang et al., 2023a; Ye et al., 2023; Wang et al., 2024a) typically employ the CLIP (Radford et al., 2021) image encoder as a substitute for the text encoder to introduce low-level details of the image. Inspired by dense reference image conditioning, recent works (Hu et al., 2023; Xu et al., 2024a) opt to train an additional reference network that uses the same initialization as the denoising network. The feature maps from the reference network are injected into the denoising network through the attention mechanism. This dual U-Net architecture significantly increases the training cost. Moreover, such dense reference image conditioning is ineffective for actions with body shape changes. Existing works neglect the fact that the keypoints of the sparse skeleton pose correspond to appearance characteristics. We argue that considering sparse skeleton pose keypoints as correspondences can provide effective appearance guidance while relaxing shape constraints.

To this end, we propose DisPose, a plug-and-play guidance module to disentangle pose guidance, which extracts robust control signals from only the skeleton pose map and reference image without additional dense inputs. Specifically, we disentangle pose guidance into motion field estimation and keypoint correspondence. First, we compute the sparse motion field using the skeleton pose. We then introduce a reference-based dense motion field to provide region-level motion signals through condition motion propagation on the reference image. To enhance appearance consistency, we extract diffusion features corresponding to key points in the reference image. These point features are transferred to the target pose by computing multi-scale point correspondences from the motion trajectory. Architecturally, we implement these disentangled control signals in a ControlNet-like (Zhang et al., 2023a) manner to integrate them into existing methods. Finally, motion fields and point embedding are injected into the latent video diffusion model resulting in accurate human image animation as shown in Figure. 1. The contribution of this paper can be summarized as:

- We propose a plug-and-play module for controllable human animation.
- We innovatively disentangle motion field guidance and keypoint correspondence from pose control to provide efficient control signals without additional dense inputs.
- Extensive qualitative and quantitative experiments demonstrate the superiority and generality of the proposed model.

## 2 RELATED WORK

**Latent Image/Video Diffusion Models.** Diffusion-based models (Ho et al., 2020; Song et al., 2020; Rombach et al., 2022; Zhang et al., 2023a; Zeng et al., 2024) have achieved remarkable success in the fields of image generation and video generation. Due to reasons such as computational intensity and information redundancy, diffusion models directly on the pixel space are hard to scale up. The latent diffusion model (LDM) (Rombach et al., 2022) introduces a technique for denoising in the latent space, which reduces the computational requirements while preserving the generation quality. In contrast to image generation, video generation requires more accurate modeling for temporal motion patterns. Recent video generation models (Blattmann et al., 2023b; Ge et al., 2023; Guo et al., 2023) utilize pre-trained image diffusion models to enhance the temporal modeling capability by inserting temporal mixing layers of various forms.

**Diffusion Models for Human Image Animation.** Recent advancements in latent diffusion models have significantly contributed to the development of human image animation. Previous human image animation models (Wang et al., 2024b; Xu et al., 2024a; Chang et al., 2023) followed the same two-stage training paradigm. In the first stage, the pose-driven image model is trained on individual video frames and corresponding pose images. In the second stage, the temporal layer is inserted to capture temporal information while keeping the image generation model frozen. Based on this stage training paradigm, Animate Anyone (Hu et al., 2023) utilizes ReferenceNet with UNet architecture to extract appearance features from reference characters. With the development of video diffusion modeling, recent work (Zhang et al., 2024) has directly fine-tuned Stable Video Diffusion (SVD) (Blattmann et al., 2023a) to replace two-stage training. To prove the effectiveness of the proposed method, we integrate DisPose on both paradigms.

**Pose Guidance in Human Image Animation.** Human image animation typically uses the skeleton pose (e.g., OpenPose (Cao et al., 2017)) as the control guide. DWpose (Yang et al., 2023) stands out as an augmented alternative to OpenPose (Cao et al., 2017) since it provides more accurate and expressive skeletons. Recent work has focused on introducing dense conditions to enhance the quality of the generated video. MagicAnimate (Xu et al., 2024a) uses DensePose (Güler et al., 2018) instead of skeleton pose, which establishes a dense correspondence between RGB images and surface-based representations. Champ (Zhu et al., 2024) renders depth maps, normal maps, and semantic maps from SMPL (Loper et al., 2015) to provide detailed pose information. However, these overly dense guidance techniques rely too much on the driving video and generate inconsistent results when the target identity is significantly different from the reference. In this paper, we propose a reference-based dense motion field that provides dense motion signals while avoiding strict geometric constraints.

## 3 PRELIMINARY

We choose Stable Diffusion (SD) as the base diffusion model in this paper since it is the most popular open-source model and has a well-developed community. SD performs the diffusion process in the latent space of a pre-trained autoencoder. The input image $I$ is transformed into a latent representation $z_0 = \mathcal{E}(I)$ using a frozen encoder $\mathcal{E}(\cdot)$. The diffusion process involves applying a variance preserving Markov process to $z_0$, where noise levels increase monotonically to generate diverse noisy latent representations:

$$z_t = \sqrt{\bar{\alpha}_t} z_0 + \sqrt{1 - \bar{\alpha}_t} \epsilon, \epsilon \sim \mathcal{N}(0, I), \tag{1}$$

where $t = 1, \cdots, T$ denotes the time steps within the Markov process, where $T$ is commonly configured to 1000, and $\bar{\alpha}_t$ represents the pre-defined noise intensity at each time step. The denoising network $\epsilon_\theta(\cdot)$ learns to reverse this process by predicting the added noise, encouraged by the mean squared error (MSE) loss:

$$\mathcal{L} = \mathbb{E}_{\mathcal{E}(I), y, \epsilon \sim \mathcal{N}(0, I), t} \left[ \| \epsilon - \epsilon_\theta \left( z_t, t, c_{\text{text}} \right) \|_2^2 \right], \tag{2}$$

where $c_{\text{text}}$ is the text embedding corresponding to $I$. The denoising network $\epsilon_\theta(\cdot)$ is typically implemented as a U-Net (Ronneberger et al., 2015) consisting of pairs of down/up sample blocks at four resolution levels, as well as a middle block. Each network block consists of ResNet (He et al., 2016), spatial self-attention layers, and cross-attention layers that introduce text conditions.

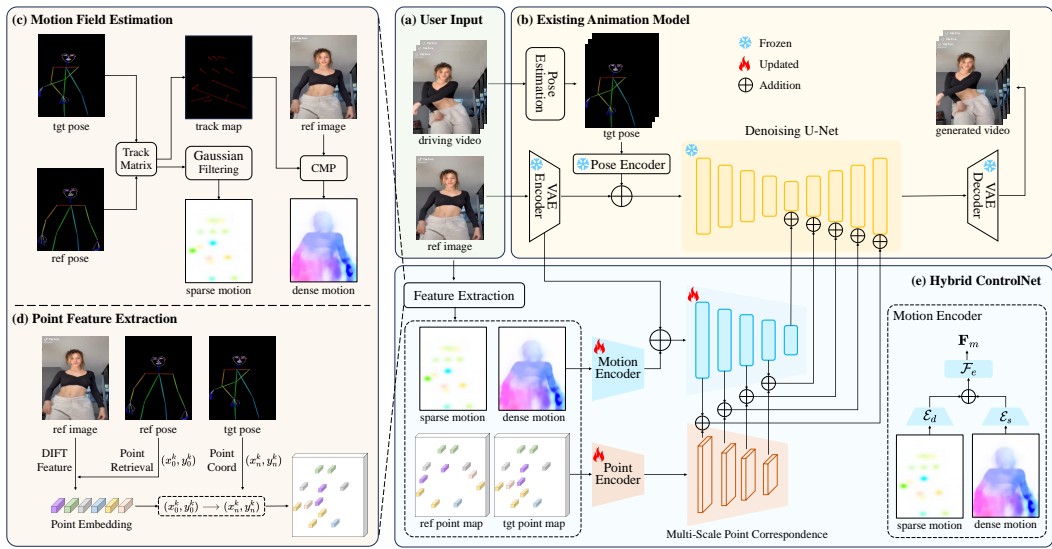

Figure 2: The overview of proposed DisPose.

# 4 DISPOSE

Given a reference image $I_{\text{ref}} \in \mathbb{R}^{3 \times H \times W}$ and a driving video $V \in \mathbb{R}^{N \times 3 \times H \times W}$. The core of our method is to disentangle efficient control guidance from only skeleton poses and reference images as shown in Figure 2, which can be applied to existing human animation methods without additional dense inputs. We first introduce sparse and dense motion field guides in Sec. 4.1. Then, we introduce reference-based keypoint correspondence in Sec. 4.2. Finally, we introduce the pipeline of hybrid ControlNet in Sec 4.3.

## 4.1 MOTION FIELD GUIDANCE

**Sparse Motion Field.** We first estimate the skeleton pose by DWpose (Yang et al., 2023) to obtain each frame's human key point coordinates. Subsequently, the key points of the reference image are used as starting points to track the motion displacement of all frames and represented as $P_{traj} = \{(x_n^k, y_n^k) \mid k = 1 \ldots K, n = 0 \ldots N\}$, where $P_{traj}$ denotes the trajectory map of the key point $k$ overall $N$ frames and $n = 0$ denotes the reference image. We calculate the track matrix $P_s$ as follows:

$$\mathbf{P}_s = \{(x_n^k - x_{n-1}^k, y_n^k - y_{n-1}^k) \mid n = 1 \ldots N\}\}, \tag{3}$$

where $K$ denotes the number of keypoints, $N$ denotes the number of frames, $\mathbf{P}_s$ denotes the trajectory map of keypoint k over all N frames, and $n = 0$ denotes the reference image. To avoid training instability caused by overly sparse trajectory matrice, we then apply Gaussian filtering to enhance $\mathbf{P}_s$ to obtain the sparse motion field $\mathbf{F}_s \in \mathbb{R}^{(N-1) \times 2 \times H \times W}$.

**Dense Motion Field.** Considering that sparse control provides limited guidance and dense control is hard to obtain during inference, we transform dense guidance into the motion propagation from the reference frame to the target pose, instead of the dense signal of the target pose. Specifically, in the inference, we reconstruct the trajectory map $\mathbf{P}_s$ as the reference-based sparse optical flow $\mathbf{P}_d$ from the reference frame to each target pose as:

$$\mathbf{P}_d = \{(x_n^k - x_0^k, y_n^k - y_0^k) \mid n = 1 \ldots N\}\}, \tag{4}$$

We then predicted the reference-based dense motion filed $\mathbf{F}_d = \text{CMP}(P_{traj}, \mathbf{P}_d, I_{ref}) \in \mathbb{R}^{(N-1) \times 2 \times H \times W}$ by condition motion propagation (CMP) (Zhan et al., 2019) based on the sparse optical flow $\mathbf{P}_d$ and the reference image $I_{ref}$. CMP (Zhan et al., 2019) is a self-supervised learning-from-motion model that acquires an image and a sparse motion field and estimates the corresponding dense motion field. Notably, the dense motion field $\mathbf{F}_d$ of each frame starts with the reference image, which avoids geometric constraints during inference.

To ensure motion estimation accuracy during training, we first extract the forward optical flow from the driving video using existing optical flow estimation models (Teed & Deng, 2020; Xu et al., 2023). We then use a watershed-based approach (Zhan et al., 2019) to sample the sparse optical flow $\mathbf{P}_d$ from the forward optical flow. See Appendix. A for details.

**Motion Encoder.** To leverage motion field as guidance, we introduce a motion encoder specifically designed for the optional flow, which includes sparse motion encoder $\mathcal{E}_s$, dense motion encoder $\mathcal{E}_d$ and feature fusion layer $\mathcal{F}_e$. $\mathcal{E}_d$ and $\mathcal{E}_s$ have the same structure and are multi-scale convolutional encoders with each stage built by `Conv-SiLU-ZeroConv` (Zhang et al., 2023a) as the basic block. The feature fusion layer $\mathcal{F}_e$ is a 2D convolution for fusing sparse motion features $\mathcal{E}_s(\mathbf{F}_s)$ and dense motion features $\mathcal{E}_d(\mathbf{F}_d)$. Finally, we compute the motion field guidance $\mathbf{F}_m$:

$$\mathbf{F}_m = \mathcal{F}_e(\mathcal{E}_s(\mathbf{F}_s) + \mathcal{E}_d(\mathbf{F}_d)) \tag{5}$$

## 4.2 KEYPOINT CORRESPONDENCE

**Point Feature Extraction.** To maintain a consistent appearance, it is crucial to correspond the content of the reference image with the motion trajectory. Specifically, we first extract the DIFT (Tang et al., 2023) features $\mathbf{D}$ of the reference image using the pre-trained image diffusion model.

Subsequently, the keypoint embedding in the reference is obtained as $\mathbf{D}(x_0^k, y_0^k)$, where $(x_0^k, y_0^k)$ is retrieved from the reference pose. Next, we initialize the keypoint correspondence map $\mathbf{F}_p$ with zero vectors and assign point embeddings according to the trajectory coordinates as:

$$f_n^{ij} = \begin{cases} \mathbf{D}(x_0^k, y_0^k), & \text{if} \quad i = x_n^k, j = y_n^k, \\ 0, & \text{otherwise.} \end{cases} \tag{6}$$

Finally, we obtain the keypoint correspondence map $\mathbf{F}_p = \{f_n \mid n = 1 \dots N\} \in \mathbb{R}^{N \times D_p \times H \times W}$ for all frames, where $D_p$ is the feature dimension of the point embedding.

**Point Encoder.** To utilize the content correspondence of key points as guidance, we generate multi-scale correspondences of sparse point feature maps and make them compatible with the U-Net encoder of the Hybrid ControlNet (Sec.4.3). We introduce the multi-scale point encoder $\mathcal{E}_p$ to maintain the key point content $\mathbf{F}_p$ from the reference image. The point encoder $\mathcal{E}_p$ consists of a series of learnable MLPs. To seamlessly integrate into existing models, we extract intermediate features of the encoder of the hybrid Controlnet. The multi-scale intermediate features of the Controlnet encoder are denoted as $\mathbf{E}_{enc}^l$, where $l$ denotes each U-Net block $l \in [1, L]$. To match the spatial size of $\mathbf{E}_{enc}^l$, we apply downsampling to the feature map between the encoder layers. We compute the multi-scale keypoint correspondence as follows:

$$\mathbf{F}_c^l = \mathcal{E}_p^l(\phi(\mathbf{F}_p, H^l, W^l)), \tag{7}$$

where $(H^l, W^l)$ are denote the spatial dimension of the $l$-th U-Net block and $\phi$ means downsampling operation. Therefore, $\mathbf{F}_c^l$ shares the same size as $\mathbf{E}_{enc}^l$. Finally, $\mathbf{F}_c$ are added elementwisely to the intermediate feature $\mathbf{E}_{enc}^l$ of the U-Net encoder as guidance: $\mathbf{E}_{enc}^l = \mathbf{E}_{enc}^l + \mathbf{F}_c^l$.

## 4.3 PLUG-AND-PLAY HYBRID CONTROLNET

After obtaining motion field guidance and keypoint correspondence, we aim to integrate these control guidance seamlessly into the U-Net architecture of existing animation models. Inspired by ControlNet (Zhang et al., 2023a), We design a hybrid ControlNet $\mathcal{F}$ to provide additional control signals for the existing animation model as shown in Figure 2(e). Specifically, given an animation diffusion model based on the U-Net architecture, we freeze its all modules while allowing the motion encoder, point encoder and hybrid ControlNet to be updated during training. Subsequently, $\mathbf{F}_m$ is added to the noise latent before being input into the hybrid ControlNet. Considering the high sparsity of the point feature $\mathbf{F}_c$, we correspondingly add $\mathbf{F}_c$ to the input of the convolutional layer. Notably, the U-Net encoder intermediate feature $\mathbf{E}_{enc}$ in Sec. 4.2 is from hybrid ControlNet rather than denoising U-Net. Finally, the control condition is computed as:

$$\boldsymbol{r} = \mathcal{F}(\boldsymbol{z}_t \mid \mathbf{F}_m, \mathbf{F}_c, t) \tag{8}$$

where $\boldsymbol{r}$ is a set of condition residuals added to the residuals for the middle and upsampling blocks in the denoising U-Net.

Table 1: Quantitative comparisons on Tiktok dataset.

| Method | VBench↑ | | | | | | FID-FVD↓ | FVD↓ | CD-FVD↓ |
|---|---|---|---|---|---|---|---|---|---|
| | Temporal Flickering | Subject Consistency | Background Consistency | Motion Smoothness | Dynamic Degree | Imaging Quality | | | |
| *Stable Diffusion1.5* | | | | | | | | | |
| MagicPose (Chang et al., 2023) | 96.65 | 95.12 | 94.55 | 98.29 | 22.70 | 63.87 | 15.53 | 1015.04 | 693.24 |
| Moore (MooreThreads, 2024) | 96.86 | 95.18 | 95.37 | 98.01 | 25.51 | 69.14 | 11.58 | 924.40 | 687.88 |
| MusePose (Tong et al., 2024) | 97.02 | 95.27 | 95.16 | 98.45 | 27.31 | 71.56 | 11.48 | 866.36 | 626.59 |
| MusePose+Ours | **97.63** | **95.70** | **95.64** | **98.51** | **31.34** | **71.89** | **11.26** | **764.00** | **622.64** |
| *Stable Video Diffusion* | | | | | | | | | |
| ControlNeXt (Peng et al., 2024) | 97.55 | 94.58 | 95.60 | 98.75 | 27.58 | 70.40 | 10.49 | 496.87 | 624.51 |
| MimicMotion (Zhang et al., 2024) | 97.56 | 94.95 | 95.36 | 98.67 | 28.42 | 68.42 | 10.50 | 598.41 | 621.90 |
| MimicMotion+Ours | **97.73** | **95.72** | **95.90** | **98.89** | **29.51** | **71.33** | **10.24** | **466.93** | **603.27** |

## 5 EXPERIMENTS

### 5.1 IMPLEMENTATIONS

**Baseline Models.** To demonstrate the effectiveness of DisPose, we integrate proposed modules into two open source human image animation models: MusePose (Tong et al., 2024) and Mimic-Motion (Zhang et al., 2024). MusePose (Tong et al., 2024) is a reimplementation of AnimateAnyone (Hu et al., 2023) by optimizing Moore-AnimateAnyone (MooreThreads, 2024), which implements most of the details of AnimateAnyone (Hu et al., 2023) and achieves comparable performance. MimicMotion (Zhang et al., 2024) is the state-of-the-art human animation model based on Stable Video Diffusion (Blattmann et al., 2023a).

**Implementation Details.** Following (Hu et al., 2023; Zhang et al., 2024), We employed DW-Pose (Yang et al., 2023) to extract the pose sequence of characters in the video and render it as pose skeleton images following OpenPose (Cao et al., 2017). We collected 3k human videos from the internet to train our model. For MusePose (Tong et al., 2024), we used `stable-diffusion-v1-5`[*] to initialize our hybrid ControlNet. We sampled 16 frames from each video and center cropped to a resolution of 512×512. Training was conducted for 20,000 steps with a batch size of 32. The learning rate was set to 1e-5. For MimicMotion (Zhang et al., 2024), we initialized our hybrid ControlNet using `stable-video-diffusion-img2vid-xt`[†]. We sampled 16 frames from each video and center crop to a resolution of 768×1024. Training was conducted for 10,000 steps with a batch size of 8. The learning rate was set to 2e-5.

**Evaluation metrics.** The video quality is evaluated by calculating the Frechet Inception Distance with Fréchet Video Distance (FID-FVD) (Balaji et al., 2019), Fréchet Video Distance (FVD) (Unterthiner et al., 2018) and Content-Debiased Fréchet Video Distance (CD-FVD) (Ge et al., 2024) between the generated video and the grounded video. Considering that these metrics are inconsistent with human judgment (Huang et al., 2024), we introduce metrics in VBench (Huang et al., 2024) to comprehensively assess the consistency of the generated video with human perception, including temporal flickering, aesthetic quality, subject consistency, background consistency, motion smoothness, dynamic degree, and imaging quality.

### 5.2 QUANTITATIVE COMPARISON

**Evaluation on TikTok dataset.** We compare our method to the state-of-the-art human image animation methods, including MagicPose (Chang et al., 2023), Moore-AnymateAnyone (MooreThreads, 2024), MusePose (Tong et al., 2024), ControlNeXt (Peng et al., 2024) and MimicMotion (Zhang et al., 2024). Following previous works (Zhang et al., 2024; Wang et al., 2024b), we use sequences 335 to 340 from the TikTok (Jafarian & Park, 2021) dataset for testing. Table 1 presents a quantitative analysis of the various methods evaluated on the TikTok dataset. The proposed methods achieve significant improvements across different baseline models. Our method achieves higher scores on VBench (Huang et al., 2024) while reducing FID-FVD and FVD scores, which indicates that the proposed method generates high-quality videos that align with human perception.

---

[*]https://huggingface.co/stable-diffusion-v1-5/stable-diffusion-v1-5.

[†]https://huggingface.co/stabilityai/stable-video-diffusion-img2vid-xt.

Table 2: Quantitative comparisons on unseen dataset.

| Method | Temporal Flickering | Subject Consistency | Background Consistency | Motion Smoothness | Dynamic Degree | Imaging Quality | Aesthetic Quality |
|---|---|---|---|---|---|---|---|
| *Stable Diffusion 1.5* | | | | | | | |
| MagicPose (Chang et al., 2023) | 92.65 | 93.71 | 98.51 | 25.67 | 63.78 | 93.65 | 46.16 |
| Moore (MooreThreads, 2024) | 92.83 | 92.42 | 98.12 | 27.43 | 65.32 | 94.61 | 47.23 |
| MusePose (Tong et al., 2024) | 93.12 | 93.97 | 98.58 | 28.72 | 65.26 | 96.41 | 49.34 |
| MusePose+Ours | **93.43** | **94.22** | **98.76** | **29.61** | **65.48** | **96.63** | **49.39** |
| *Stable Video Diffusion* | | | | | | | |
| ControlNeXt (Peng et al., 2024) | 93.25 | 94.27 | 98.70 | 28.42 | 64.36 | 97.42 | 49.10 |
| MimicMotion (Zhang et al., 2024) | 93.32 | 94.12 | 98.50 | 29.81 | 64.51 | 97.45 | 49.03 |
| MimicMotion+Ours | **93.59** | **94.35** | **98.75** | **30.02** | **65.56** | **97.80** | **49.93** |

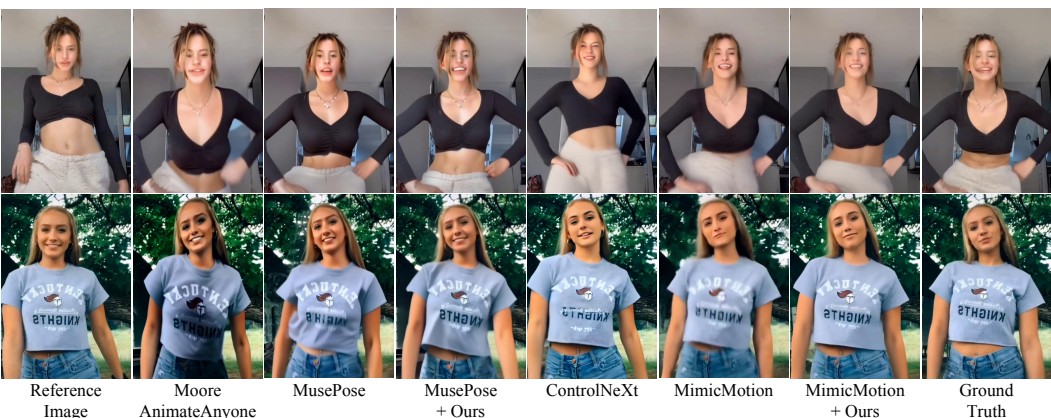

| Reference Image | Moore AnimateAnyone | MusePose | MusePose + Ours | ControlNeXt | MimicMotion | MimicMotion + Ours | Ground Truth |

Figure 3: Qualitative comparisons between our method and the state-of-the-art models on the TikTok dataset.

**Evaluation on unseen dataset.** Training videos collected from the Internet may exhibit domain proximity with the TikTok (Jafarian & Park, 2021) test set. We construct an unseen dataset to further compare the generalizability of various methods. We collect 30 high-quality human videos and generate reference images with diverse styles using InstanID (Wang et al., 2024a). Due to the unavailability of the ground truth corresponding to the generated reference images, we use VBench (Huang et al., 2024) as the quantitative metric as shown in Table 2.

## 5.3 QUALITATIVE RESULTS

**Comparison with state-of-the-art methods.** Figure 3 illustrates the qualitative results between the various models on the TikTok dataset. Thanks to the motion field guidance and keypoint correspondence, our method can produce reasonable results with significant pose variation.

**Comparison with dense condition.** To compare the proposed method with the existing dense condition, we conduct qualitative experiments in Figure 4. Champ (Zhu et al., 2024) represents human body geometry and motion features through rendered depth images, normal maps, and semantic maps obtained from SMPL (Loper et al., 2015) sequences. Since rendering an accurate human body model for an arbitrary reference character during inference is virtually impossible, Champ achieves rough shape alignment by the parametric human model. This leads to dense conditional distortions in some human body regions (e.g., face and hands) thus degrading the video quality. Moreover, parametric alignment may fail when there are significant differences in the shape and layout between the reference image and the driving video resulting in erroneous results as shown in the last case in Figure 4. In contrast to the previous dense condition, we introduce a reference-based dense motion field through the motion propagation of the skeleton pose as shown in Figure 5, which provides dense signals while avoiding the strict constraints of the target pose.

**Cross-identity animation.** Beyond animating each reference character with the corresponding motion sequence, we further investigate the cross-identity animation capability of DisPose as shown in

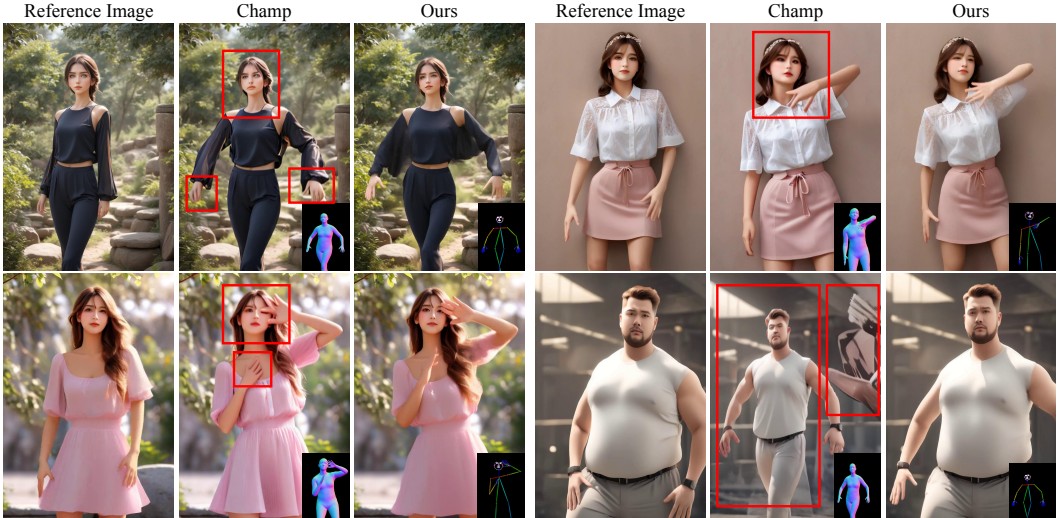

| Reference Image | Champ | Ours | Reference Image | Champ | Ours |

Figure 4: Qualitative comparison of our approach with the dense control-based method.

Table 3: Ablation study on different control guidance. "w/o Motion" denotes the model configuration that disregards motion filed guidance. "w/o Point" indicates the variant model that removes the keypoint correspondence.

| Method | VBench↑ | | | | | | FID-FVD↓ | FVD↓ |
|---|---|---|---|---|---|---|---|---|
| | Temporal Flickering | Subject Consistency | Background Consistency | Motion Smoothness | Dynamic Degree | Imaging Quality | | |
| w/o Motion | 97.66 | 95.04 | 95.31 | 98.75 | 29.42 | 69.53 | 10.31 | 478.91 |
| w/o Point | 97.47 | 95.57 | 95.43 | 98.42 | 29.14 | 70.14 | 10.28 | 498.74 |
| Full Model | **97.73** | **95.72** | **95.90** | **98.89** | **29.51** | **71.33** | **10.24** | **466.93** |

Figure 6. Our method generates high-quality animations for the reference image that are faithful to the target motion, proving its robustness. See Appendix. B for more qualitative results.

## 5.4 ABLATION STUDY

**Quantitative results.** As shown in Table 3, the full configuration of the proposed method outperforms the other variants in all metrics. The motion field guidance provides region-level control signals that enhance video consistency, resulting in lower FID-FVD and FVD. The keypoint correspondence creates the feature map of the target pose by localizing the semantic point features of the reference image, which makes the generated video more consistent with human perception.

**Semantic correspondence.** To better understand the performance of keypoint correspondences, we visualize the semantic correspondences of the variant models in Figure 7. Specifically, we select a human region (e.g., hand) from the source image and query the target image using the corresponding DIFT features. The keypoint correspondence can localize the correct semantic region from the various characters.

## 6 CONCLUSION

In this paper, we present DisPose, a plug-and-play module for improving human image animation, which aims to provide efficient conditional control without additional dense inputs. To achieve this, we disentangle pose control into motion field guidance and keypoint correspondence. To obtain the motion field guidance, we first construct the tracking matrix from the skeleton pose, and then obtain the sparse and dense motion fields by Gaussian filtering and conditional motion diffusion, respectively. Moreover, we introduce the keypoint correspondence of diffusion features to explore the semantic correspondence in image animation. Finally, we integrate the extracted guidance features into a hybrid control network. Once trained, our model can be integrated into existing human

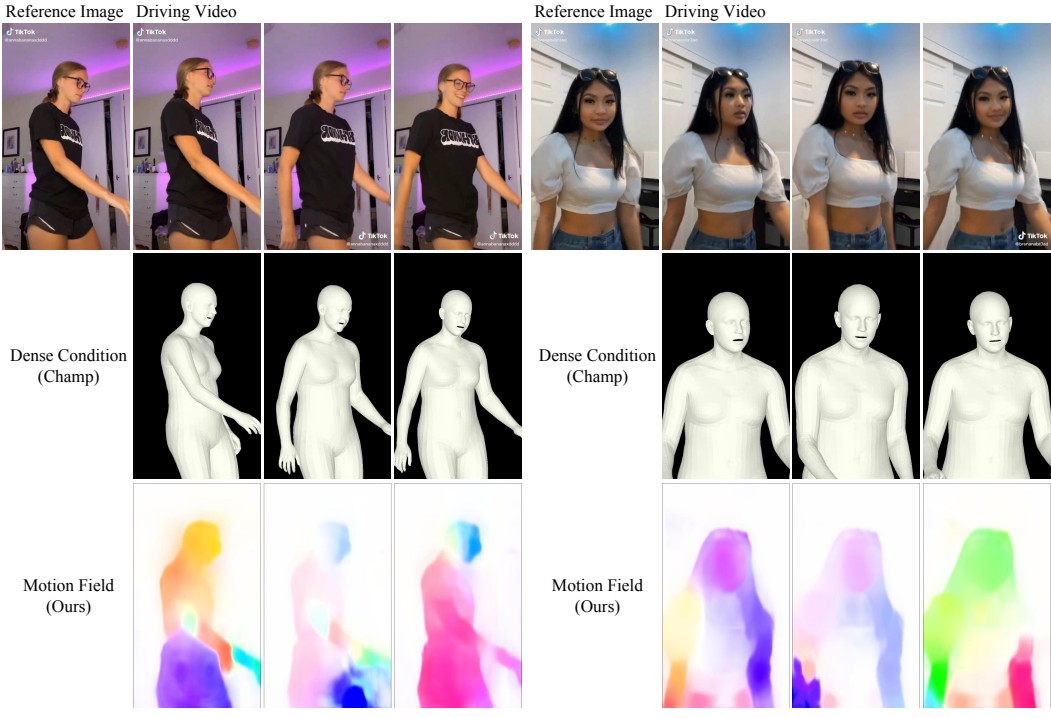

Figure 5: Comparison of our reference-based dense motion field and existing dense conditions.

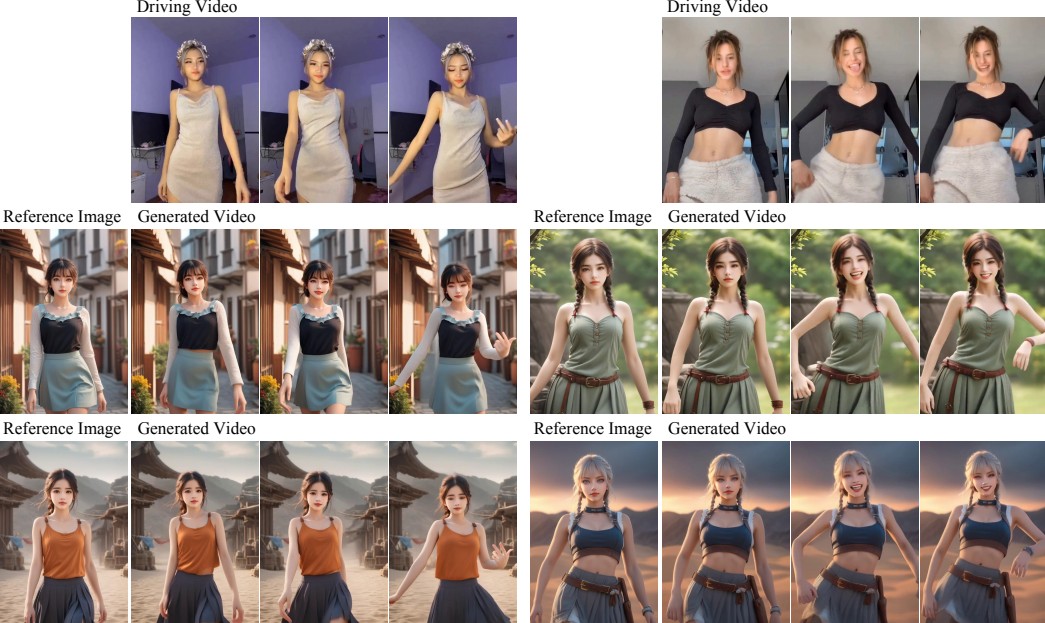

Figure 6: The demonstration of cross ID animation from the proposed method.

image animation models. Extensive evaluation of various models also validates the effectiveness and generalizability of our DisPose.

# 7 LIMITATIONS AND FUTURE WORKS

Despite our DisPose enhances motion guidance and appearance alignment, the ability to synthesize unseen parts for characters is still limited. As shown in Figure 8, we attempt to generate multi-view

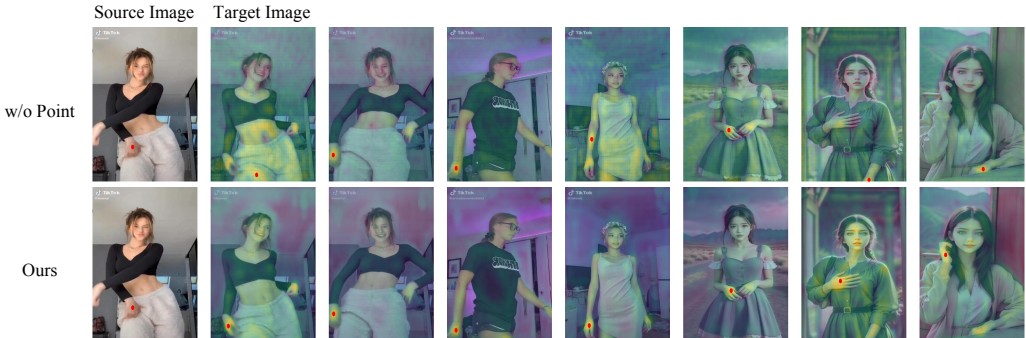

Figure 7: Qualitative analysis of semantic correspondence. Given a red source point in an image (far left), we use its diffusion feature to retrieve the corresponding point in the image on the right.

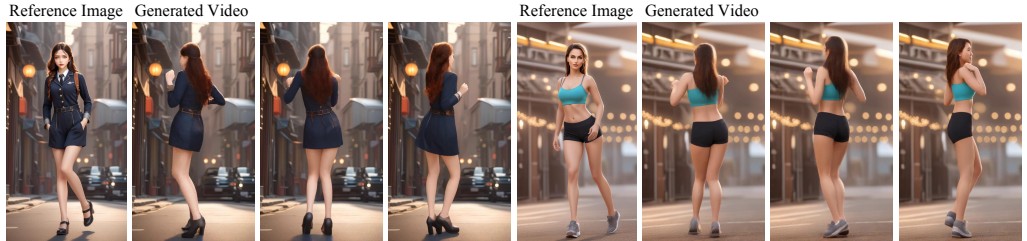

Figure 8: Qualitative results of our method for multi-view video generation.

videos for the single-view reference image. In the future, we will explore camera control or multi-view synthesis models for capturing multi-view reference information. Moreover, introducing the 3D sparse pose as a control condition can also address this issue.

## 8 ETHICS STATEMENT

We clarify that all characters in this paper are fictional except for the TikTok (Jafarian & Park, 2021) dataset. We strongly condemn the misuse of generative artificial intelligence to create content that harms individuals or spreads misinformation. However, we acknowledge the potential for misuse of our approach. This is because it focuses on human-centered animation generation. We uphold the highest ethical standards in our research, including adherence to legal frameworks, respect for privacy, and encouragement to generate positive content.

## 9 ACKNOWLEDGEMENTS

This work was supported by the Hong Kong SAR RGC Early Career Scheme (26208924), the National Natural Science Foundation of China Young Scholar Fund (62402408), Huawei Gift Fund, and the HKUST Sports Science and Technology Research Grant (SSTRG24EG04).

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

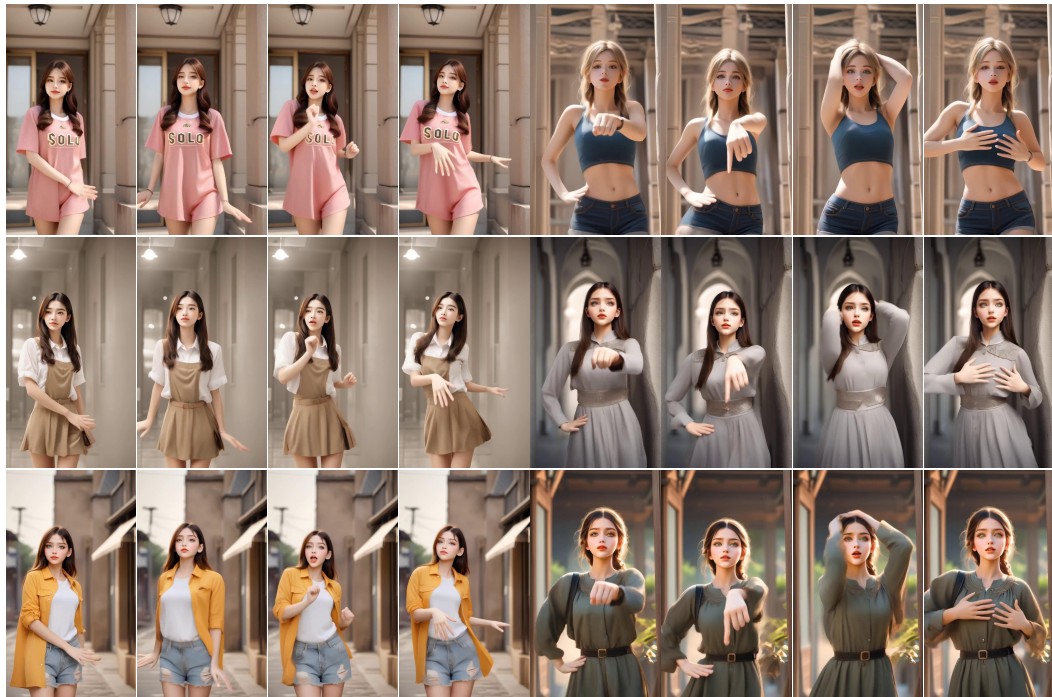

Figure 9: More Qualitative Comparisons.

## A  SAMPLING FROM WATERSHED

During training, the sparse optical flow $\mathbf{P}_d$ is sampled from the target optical flow. For effective propagation, those guidance vectors should be placed at some keypoints where the motions are representative. We adopt a watershed-based (Zhan et al., 2019) method to sample such keypoints. Given the optical flow of an image, we first extract motion edges using a Sobel filter. Then we assign each pixel a value to be the distance to its nearest edge, resulting in the topological-distance watershed map. Finally, we apply Non-Maximum Suppression (NMS) with kernel size $K_f$ on the watershed map to obtain the keypoints. We can adjust $K_f$ to control the average number of sampled points. A larger $K_f$ results in sparser samples. Points on image borders are removed. With the watershed sampling strategy, all the keypoints are roughly distributed on the moving objects.

## B  MORE QUANTITATIVE RESULTS

Figures 9 and Figure 10 illustrate more qualitative results.

## C  MORE ABLATION ANALYSES

As shown in Figure 11, the region-level guidance provided by our motion field guidance facilitates the enhancement of consistency across body regions. The proposed keypoints correspondence improves generation quality by aligning DIFT features of the skeleton pose, e.g., facial consistency.

## D  MORE DETAILS OF MOTION FIELD GUIDANCE

There is a gap between the inference and the training optical flow. (1) During inference, we do not propose extracting the forward optical flow directly from the driving video, as it ignores the gap between the reference character and the driving video. As shown in Figure 14(a)and Figure 15(a), directly using the forward optical flow as motion guidance is clearly inconsistent with the reference image. (2) When there is a large difference between the reference image and the driving video, it is

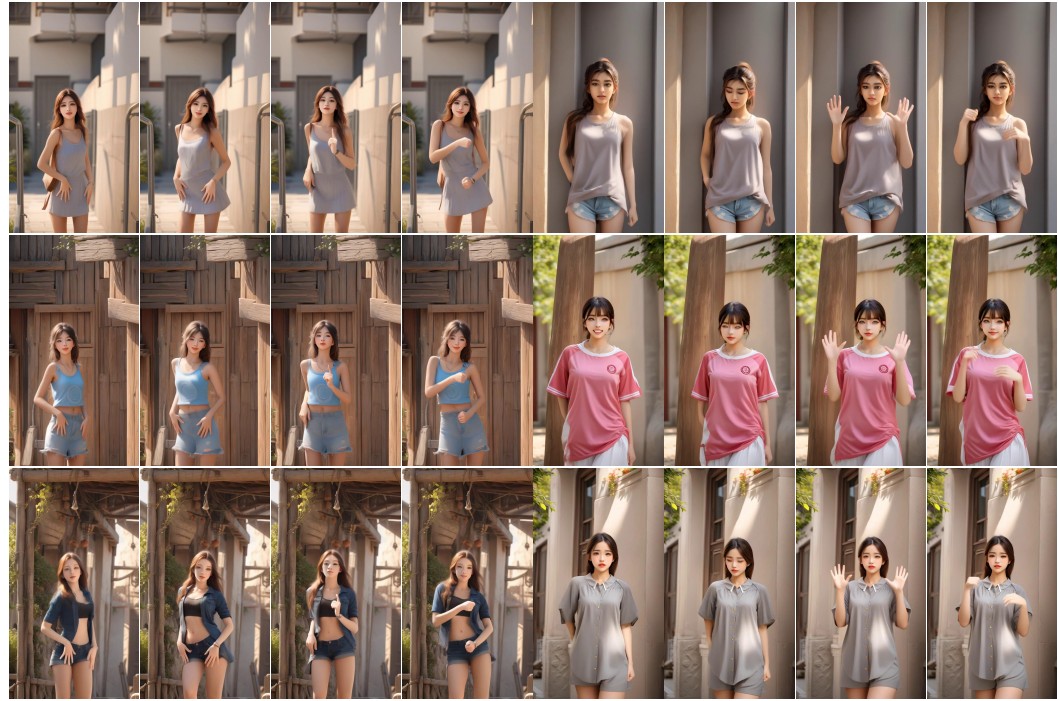

Figure 10: More Qualitative Comparisons.

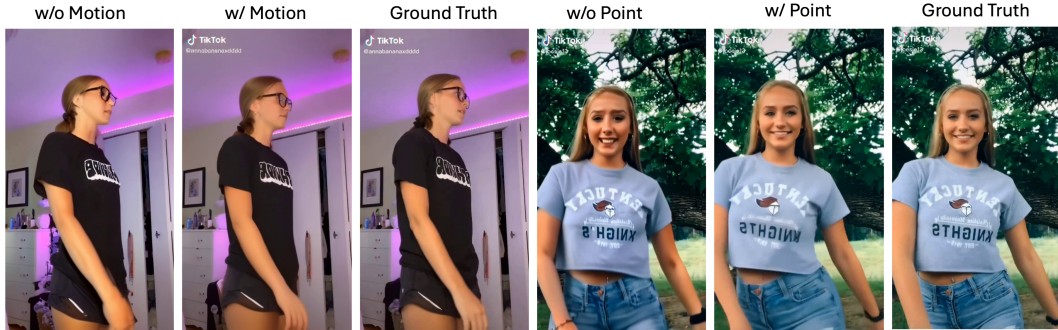

Figure 11: Qualitative results about motion field guidance and keypoints correspondence.

impossible to get the corresponding motion field by the existing optical flow estimation model, as shown in Figure 15(b). Therefore, we have to compute Pd differently during inference.

Although the dense motion field we proposed in Section 4.1 can adapt to different body variations during inference. However, there are two limitations of this dense motion field: (1) there is a gap with the motion field extracted from real videos, and (2) the low computational efficiency is not suitable for use during training. Considering that pairs of training data have no body changes, to utilize accurate control signals during training and to improve computational efficiency, we approximate the optical flow during inference by sampling sparse optical flow before prediction as shown in Figure 14(c).

# E MORE ABLATION STUDY

## E.1 THE IMPACT OF HYBRID CONTROLNET ARCHITECTURE

We show the impact of hybrid ControlNet architecture in Table 4. Specifically, we design two variant architectures, (1) Exp1: inserting the motion field into the denoising network instead of the hybrid

Table 4: The impact of hybrid ControlNet.

| Methods | FID-FVD↓ | FVD↓ |
|---|---|---|
| Exp1 | 10.43 | 514.83 |
| Exp2 | 10.94 | 551.32 |
| Full Model | 10.24 | 466.93 |

Table 5: The impact of CMP.

| Methods | subject consistency↑ | background consistency↑ |
|---|---|---|
| Full Model w/o CMP | 93.94 | 97.83 |
| Full Model | 94.35 | 98.75 |

Table 6: Performance comparisons for image-level metrics.

| Methods | SSIM ↑ | PSNR↑ | LPIPS↓ | L1↓ |
|---|---|---|---|---|
| MusePose | 0.788 | 19.14 | 0.263 | 2.46E-05 |
| MusePose+Ours | 0.811 | 19.36 | 0.238 | 2.26E-05 |
| MimicMotion | 0.749 | 18.32 | 0.272 | 2.71E-05 |
| MimicMotion+Ours | 0.781 | 19.58 | 0.242 | 2.42E-05 |

controller as shown in Figure 12(a), and (2) Exp2: removing the hybrid ControlNet and inserting the motion field guidance and keypoint correspondence into the denoising network as shown in Figure 12(b). Exp1 shows that the motion field needs to be jointly optimized with U-Net to provide the correct representation. Exp2 shows that complex motion information and appearance features cannot be modeled with only two shallow encoders.

### E.2 THE IMPACT OF CMP.

We provide the ablation analysis of CMP in Table 5, which shows that CMP can improve the consistency of the generated video.

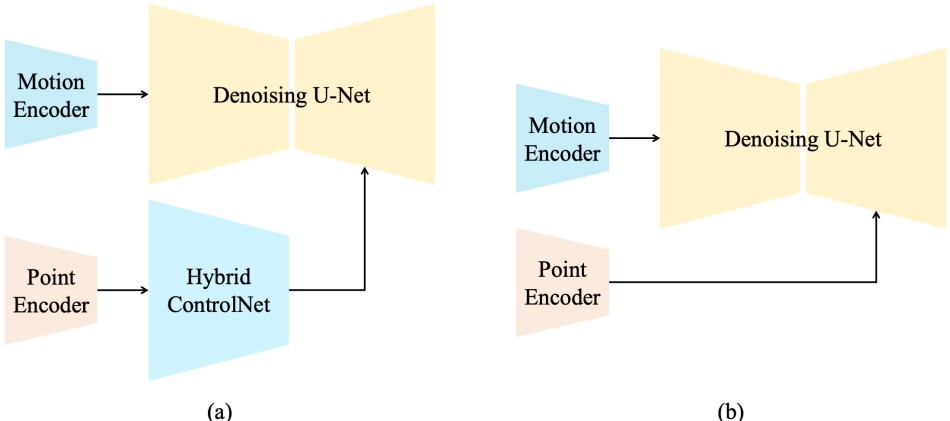

(a)                                     (b)

Figure 12: Different hybrid ControlNet architectures.

## F MORE PERFORMANCE COMPARISONS.

To further evaluate the generated results, we provide performance comparisons for image-level metrics in Table 6. Compared to the baseline model, our method achieves significant improvements in all metrics.

## G TRAINABLE PARAMETERS AND INFERENCE TIME

We compare the trainable parameters and inference time of the different models in Table 7. For a fair comparison, the size of the generated video is set to 576x1024. Our method requires fewer trainable

Table 7: Performance comparisons for image-level metrics.

| Methods | trainable parameters(MB) | infer time (sec/frame) |
|---|---|---|
| MusePose | 2072.64 | 3.37 |
| MimicMotion | 1454.19 | 1.61 |
| MimicMotion+Ours | 653.40 | 2.36 |

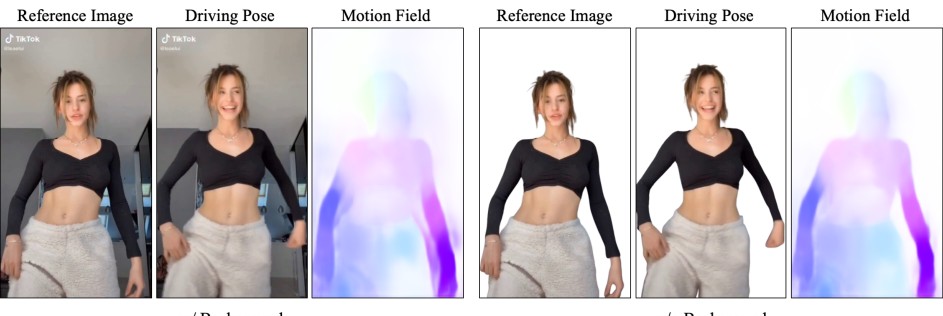

Figure 13: Analysis of background noise.

parameters based on the baseline model. During inference, our method estimates the motion field for the reference image, which increases inference time a little.

## H   ANALYSIS OF BACKGROUND NOISE.

Since our motion fields are not extracted directly from the driving video, some noise due to estimation errors may be introduced. As shown in Figure 13, the motion field of the reference image without the background is more accurate than the complex background.

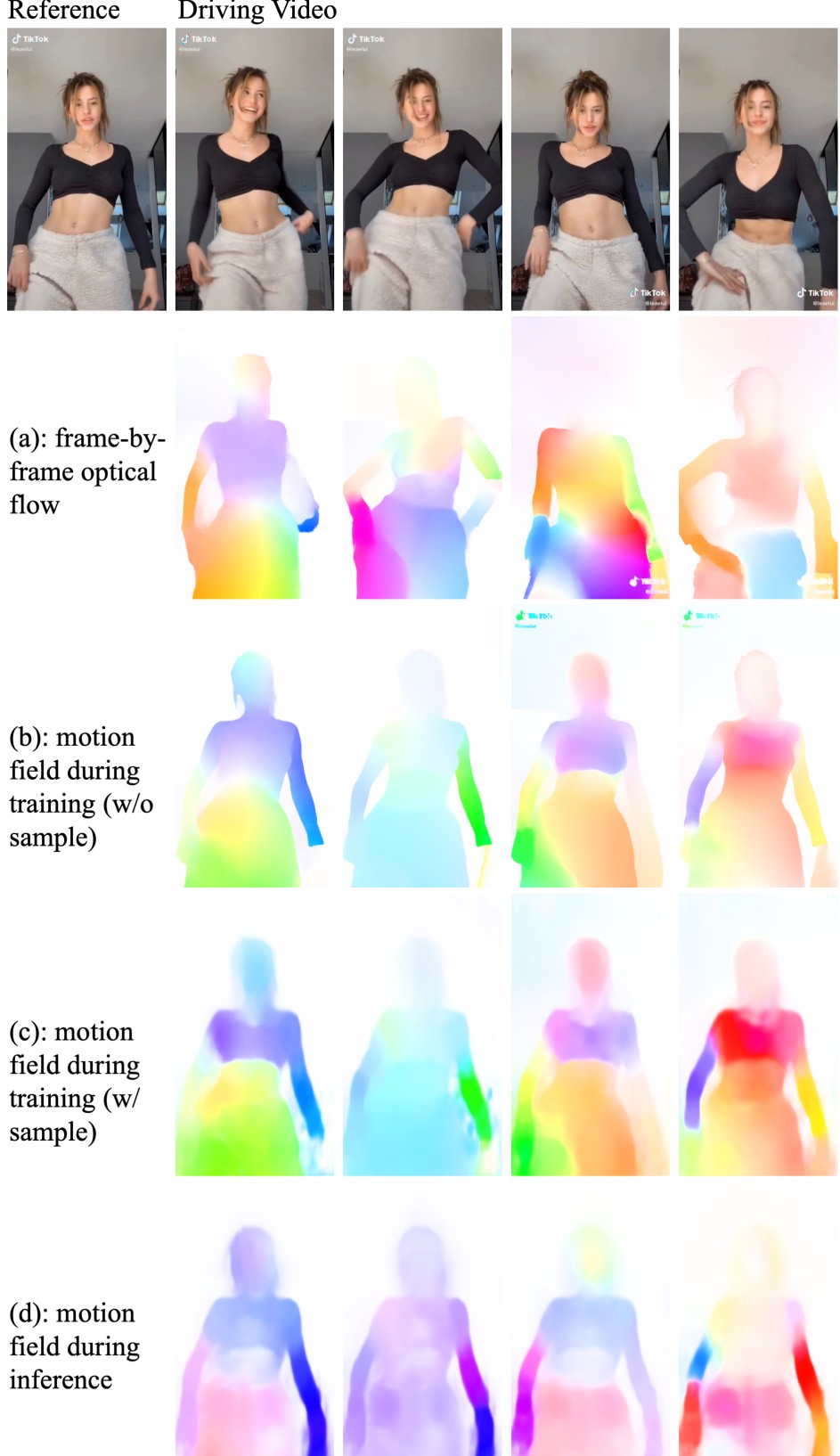

Figure 14: Body matched motion field visualization.

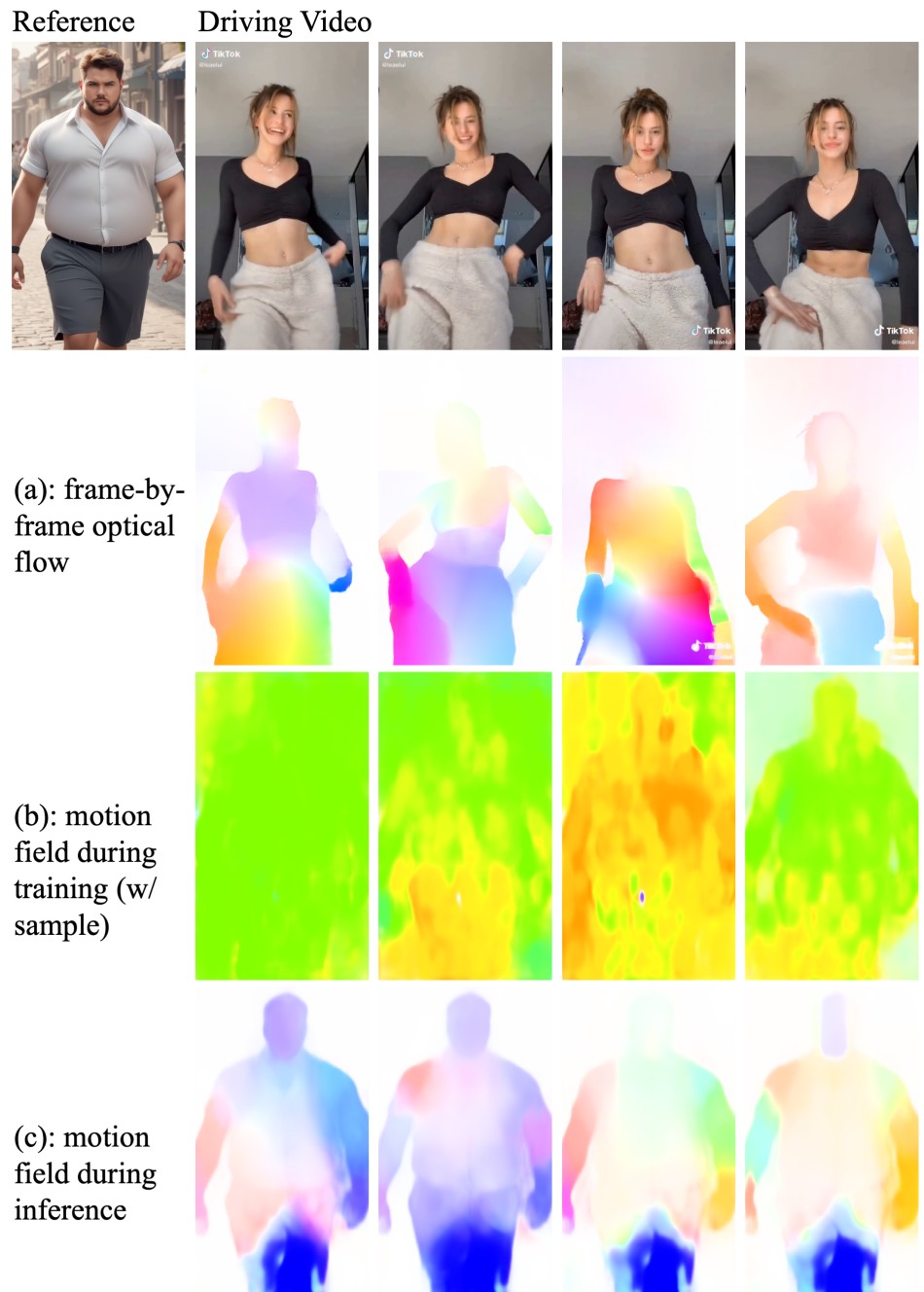

Figure 15: Body mismatched motion field visualization.

