# OpenReview forum: "DisPose: Disentangling Pose Guidance for Controllable Human Image Animation"
_ICLR.cc/2025/Conference — ICLR 2025 Poster_

### Official Review · Reviewer_c9da · 2024-10-26

**Soundness:** 3
**Presentation:** 3
**Contribution:** 3
**Rating:** 8
**Confidence:** 5

**Summary:**

This paper proposed DisPose, a human pose retargeting framework with high-quality motion control. The core of the proposed method is to disentangle control guidance from skeleton poses and reference images.

**Strengths:**

The paper is well-written and easy to understand. The motivation for extracting sparse and dense motion fields makes much sense for this task and the DIFT feature provides content correspondence of key points as guidance. The plug-in module follows the ControlNet architecture and provides flexibility for using the pre-trained weight for other customized SD-UNet. I found Figure 7 extremely helpful for understanding the keypoint correspondences.

**Weaknesses:**

1. Is there any discussion or comparison of the efficiency, e.g., trainable parameters and inference time for a single batch?

2. Metric for video generation evaluation. I understand the authors follow previous works and adopt FVD as the video evaluation metric. However, this metric has recently been widely criticized by the community because of its inaccuracy in reflecting the overall quality. I wonder what the performance comparison would be if debiased FVD is used for evaluation. From [here](https://content-debiased-fvd.github.io/)

3. I appreciate the authors for providing video visualizations in the supp. Are there any side-by-side video comparisons between this work and recent baselines? It would be better to judge the temporal consistency of the video quality and the effectiveness of motion control.

4. How does the model generalize to out-of-domain real human identities? E.g. Old people?

5. (Minor) For the input of the reference image, did the authors utilize ReferenceNet for the SD-1.5-based model and the same setting of SVD for comparison to MimicMotion?

6. (Minor) L78 There's a mistake in the cited works with an additional reference network. DisCo didn't adopt this architecture but MagicPose and Champ did.

I'm more than **willing to raise** my score if my concerns are addressed.

**Questions:**

Please see the weakness section.

---

> ### Author Response · Authors · 2024-11-21
> **Official Response to Reviewer c9da**
>
> We greatly appreciate your valuable feedback. Therefore, we have incorporated several additional experiments into the paper to showcase the effectiveness of DisPose. These newly added experiments are highlighted in $\color{red}{red}$ within the revised version of the paper.
>
> >**Q1: Trainable parameters and inference time.**
>
> We compare the trainable parameters and inference time of the different models in the table below. For a fair comparison, the size of the generated video is set to 576x1024. Our method requires fewer trainable parameters based on the baseline model.
> During inference, our method estimates the motion field for the reference image, which increases inference time a little.
> We have updated these results in Table 7 of the manuscript.
>
> |Methods  |trainable parameters(MB) |infer time (sec/frame) |
> |:---                |:---:  |:---:|
> |MusePose	|2072.64	|3.37 |
> |MimicMotion	|1454.19	|1.61 |
> |MimicMotion+Ours	|653.4	|2.36|
>
> >**Q2: The performance comparison on CD-FVD.**
>
> We show the performance comparison on CD-FVD[1] using VideoMAE-v2-SSv2 features in the following table. We have updated these results in Table 1 of the manuscript.
>
> |Model	|CD-FVD|
> |:---                |:---:|
> |MagicPose|	693.24|
> |Moore	|687.88|
> |MusePose|	626.59|
> |MusePose+Ours|	622.64|
> |ControlNeXt	|624.51|
> |MimicMotion	|621.90|
> |MimicMotion+Ours|	603.27|
>
> >**Q3: More side-by-side video comparisons.**
>
> We provide side-by-side video for cross-pose and cross-identity on the project page (https://anonymous.4open.science/r/DisPose-AB1D). Compared to the baseline model, our method improves overall quality, background consistency, and character consistency.
>
> >**Q4: The out-of-domain real human identity, e.g. old people.**
>
> 1. We provide an out-of-domain real human identity comparison for the elderly on our project page. Notably, since our method is trained with fewer parameters using fewer videos, our out-of-domain generalization ability is also affected by the baseline model.
>
> 2. For the generalization of facial identity, face-swapping[2] as a post-processing is an efficient method.
>
> 3. Moreover, extracting identity features from existing personalization generation[3,4] methods and then inserting them into existing models can also improve out-of-domain generalization.
>
> [1] On the Content Bias in Fréchet Video Distance. CVPR2024.
>
> [2] https://github.com/s0md3v/roop
>
> [3] Instantid: Zero-shot identity-preserving generation in seconds. 2024.
>
> [4] Ip-adapter: Text compatible image prompt adapter for text-to-image diffusion models. 2023.
>
> >**Q5: Settings for comparison with MimicMotion.**
>
> MimicMotion and ControlNeXt are trained based on SVD and do not use ReferenceNet. we used exactly the same settings when comparing them. To the best of our knowledge, training ReferenceNet for SVD-based models would be beneficial to preserve a consistent appearance, but it would increase the model parameters and training costs.
>
> >**Q6: Incorrect citation in L78.**
>
> Apologies for our oversight, we have updated the manuscript.

---

> > ### Comment · Reviewer_c9da · 2024-11-23
> > **Response to rebuttal**
> >
> > I appreciate the authors' effort in providing additional video visualizations of out-of-domain identities and additional CD-FVD experiments. My concerns have been addressed. Hence I'm raising my score to accept.

---

> > > ### Author Response · Authors · 2024-11-25
> > >
> > > Thank you very much for your thoughtful feedback and for raising your score. Your suggestions are invaluable in helping us improve the paper. We sincerely appreciate your recognition of our efforts to address the major concerns.

---

### Official Review · Reviewer_PDfA · 2024-10-27

**Soundness:** 3
**Presentation:** 3
**Contribution:** 2
**Rating:** 6
**Confidence:** 4

**Summary:**

This paper proposes a plug-and-play method to disentangle pose guidance, which extracts robust control signals from only the skeleton pose map and reference image without additional dense inputs. The proposed method generates a dense motion field from a sparse motion field and the reference image, and extracts diffusion features corresponding to pose keypoints from the reference image. Extensive qualitative and quantitative experiments validate the effectiveness of the proposed method.

**Strengths:**

1. The topic of controllable human animation is valuable and the control signals without additional dense input is promising to improve the generalizability of generation model.

2. The authors have conducted extensive and thorough experiments of different baselines on several datasets to validate the effectiveness of the proposed method.

**Weaknesses:**

1.	The key idea of the paper is to disentangle motion field guidance and keypoint correspondence from pose control. However, the authors just combine motion field and keypoint extraction to empower the generation ability of the original animation diffusion model. The detailed disentanglement operation is missing.

2.	For the dense motion field of Section 4.1, the authors choose to sample the sparse optical flow Pd from the forward optical flow obtained from existing optical flow estimation models, and then predict the dense motion field using CMP. However, why not directly use the dense motion field obtained by existing methods? Please give a detailed explanation of the reasons for sampling before predicting and the benefits of this way. In addition, why do authors calculate Pd differently for training and inference?

3.	The point correspondence is extracted according to the keypoint coordinates of pose and how can this kind of correspondence be helpful to maintain a consistent appearance? The authors are suggested to provide more ablation study results and visual comparison results to validate the effectiveness of multi-scale point correspondence.

4.	For the part of Point Encoder of Section 4.2, the descriptions of the U-Net encoder and multi-scale intermediate features are confusing. The authors are suggested to emphasize that the intermediate feature is from hybrid ControlNet in this part.

5.	The reference image used of the paper is realistic, however, the provided demo video is a little cartoon-style and not realistic enough.

**Questions:**

Please refer to the questions and suggestions in the “Weaknesses” part.

---

> ### Author Response · Authors · 2024-11-21
> **Official Response to Reviewer PDfA**
>
> We greatly appreciate your valuable feedback. Therefore, we have incorporated several additional experiments into the paper to showcase the effectiveness of DisPose. These newly added experiments are highlighted in $\color{red}{red}$ within the revised version of the paper.
>
> > **Q1: The detailed disentanglement operation.**
>
> 1. The motion field guidance and key point correspondences are not input by the user, rather we disentangle them from the user's input of the driving pose.
>
> 2. As shown in Figure 2(c)(d) of the manuscript, the detailed operation consists of extracting the motion fields and keypoint correspondences.
>
> 3. Section 4.1 introduces the disentanglement operations for sparse and dense motion fields. Specifically, we first estimate the skeleton pose by DWpose. Then, we compute the trajectory matrix (Ps) and apply Gaussian filtering to obtain the sparse motion field (Fs). Next, we predict the reference-based dense motion field (Fd) via conditional motion propagation (CMP).
>
> 4. Section 4.2 presents the detailed operation of keypoint feature extraction. Specifically, we first extract the DIFT features of the reference image using the pre-trained image diffusion model and then obtain the keypoint embedding according to the keypoint coordinates in the reference pose. We initialize the keypoint correspondence map (Fp) according to the trajectory coordinates of the driving pose.
>
> > **Q2: Reasons for differences in the calculation of Pd for training and inference.**
>
> There is a gap between the inference and the training optical flow.
> 1. During inference, we do not propose extracting the forward optical flow directly from the driving video, as it ignores the gap between the reference character and the driving video. As shown in Figure 13(a) and Figure 14(a) of Appendix D, directly using the forward optical flow as motion guidance is clearly inconsistent with the reference image.
> 2. When there is a large difference between the reference image and the driving video, it is impossible to get the corresponding motion field by the existing optical flow estimation model, as shown in Figure 14(b) of Appendix D. Therefore, we have to compute Pd differently during inference.
>
> >**Q3: Reasons for sampling before prediction.**
>
> Although the dense motion field we proposed in Section 4.1 can adapt to different body variations during inference. However, there are two limitations of this dense motion field: (1) there is a gap with the motion field extracted from real videos, and (2) the low computational efficiency is not suitable for use during training. Considering that pairs of training data have no body changes, to utilize accurate control signals during training and to improve computational efficiency, we approximate the optical flow during inference by sampling sparse optical flow before prediction, as shown in Figure 13(c) of Appendix D.
>
> > **Q4: The effectiveness of multi-scale point correspondence.**
>
> 1. First of all, the key points of the human body correspond to the most distinctive appearance characteristics. We capture the appearance characteristics of the reference character by extracting the diffusion features corresponding to the keypoint coordinates in the reference image. Notably, our keypoint correspondence is not just a set of coordinates; it is a set of feature maps that represent key appearance information.
>
> 2. Moreover, Figure 7 in the manuscript shows that keypoint correspondence can localize to the correct semantic region during cross-pose and cross-identity, which justifies that keypoint correspondence can provide information about appearance content. Table 3 in the manuscript demonstrates that removing keypoint correspondences affects various metrics.
>
> 3. To move a step further, we provide qualitative experiments in Figure 11 of Appendix D, which prove that keypoint correspondence contributes to appearance consistency.
>
> > **Q5: Description of the Point Encoder from Section 4.2.**
>
> To seamlessly integrate into existing models, we introduce keypoint correspondences into the intermediate features of the hybrid control network encoder. Thanks to your suggestion, we have updated Section 4.2 of the manuscript.
>
> >**Q6: More realistic generation results.**
>
> As discussed in Section 8 of the manuscript, we provide the qualitative results of the synthetic characters to avoid leakage of personal privacy. Thanks for your thoughtful comments. We provide real-life results on the project page (https://anonymous.4open.science/r/DisPose-AB1D). Please do not distribute or divulge these personal portraits.

---

> > ### Author Response · Authors · 2024-11-26
> >
> > Dear reviewer:
> >
> > As the discussion period is nearing its end, we hope to know whether our response has addressed your concerns to merit an increase in the rating, or if there are any issues that you would like us to clarify.
> >
> > Thank you once again for your time and effort.
> >
> > Sincerely
> > Authors of Paper 6301

---

> > > ### Author Response · Authors · 2024-12-01
> > >
> > > Dear Reviewer PDfA,
> > >
> > > We hope this email finds you well. With the discussion period coming to a close, We wanted to follow up and see if there are any additional questions or concerns about our paper or the rebuttal we provided earlier that we could help clarify.
> > >
> > > Your detailed feedback has been Insightful, and we have put significant effort into addressing the points you raised. If there are any remaining aspects where further clarification might strengthen your understanding of our work, we would be willing to provide more information.
> > >
> > > Furthermore, if you feel our responses have addressed your concerns effectively, we would greatly appreciate it if you would consider revisiting your initial rating of our submission. Your expert evaluation plays a crucial role in shaping the final outcome, and we sincerely appreciate your time and efforts throughout this review process.
> > >
> > > Thank you again for your dedication to improving the quality of submissions. Please feel free to let us know if there is anything else we can assist with.
> > >
> > > Best regards,
> > >
> > > Authors of Submission 6301

---

> > > > ### Comment · Area_Chair_KQy4 · 2024-12-01
> > > >
> > > > Dear Reviewer PDfA,
> > > >
> > > > It would be helpful to provide feedback on the authors' responses to indicate any remaining issues.
> > > >
> > > > Thank you.
> > > >
> > > > AC

---

> > > > > ### Comment · Reviewer_PDfA · 2024-12-02
> > > > >
> > > > > The authors have addressed my main conern of the proposed method and provided the detailed explanation of motion disentanglement and the calculation of sparse optical flow. The additional real-life results also demonstrate the superior of the proposed method. Therefore, I decide to raise my rating and am inclined to accept the paper.

---

### Official Review · Reviewer_ZwXk · 2024-11-02

**Soundness:** 2
**Presentation:** 3
**Contribution:** 2
**Rating:** 6
**Confidence:** 3

**Summary:**

The paper presents DisPose, a ControlNet utilizing optical flow/motion fields to condition Latent Diffusion Models (LDM) for image animation and motion retargeting. Instead of using dense, template-based body pose signals like SMPL and OpenPose, DisPose leverages sparse keypoints and dense optical flow fields to better handle body shape variations between the reference and the target.

Experimental results show improvement over prior work on VBench metrics.

**Strengths:**

The paper has the following strength:
- Simple, intuitive design to disentangle motion representation, and achieves better synthesis quality and appearance consistency.
- Plug-and-play potential: DisPose, when combining with previous approaches, improves the performance on the evaluated metrics.

Overall, the paper is well-written and easy to follow.

**Weaknesses:**

The paper has the following weaknesses:
- Lack of diversity in qualitative results. Both the figures, and the supp. videos focus mainly on re-animating generated/synthetic characters. These synthetic characters often have simplified, smoothed-out details, with very similar appearances and aesthetics. It is therefore unclear how the proposed method works on enhancing/improving realistic details. Particularly, since the paper already includes TikTok videos (that have real humans) for training and evaluation, it would be great if the paper can include those results.
- Using optical flow is a good idea, but it may only work for static background. For noisy background, such as swaying trees and leaves, the motion field would likely capture those motion, and thus the model may incorrectly associate these background noises with the input motion.
- Evaluation section can be further improved. It makes sense to use VBench due to the generative nature of DisPose. But at the same time, it is possible to check the appearance consistency by using the TikTok videos. One can split the video for the same subject into training and testing part, and examine whether DisPose/baselines can transfer the testing pose while preserving the subject identity.

**Questions:**

My main concerns regarding this submission is the rigorousness of the qualitative and quantitative results, as stated above in the weakness section.

Please discuss these issues accordingly, and I will update my scores accordingly if the problems are sufficiently addressed or discussed. Note that the method/DisPose does not need to be perfect, but it is important to provide a comprehensive view of the proposed method.

---

> ### Author Response · Authors · 2024-11-21
> **Official Response to Reviewer ZwXk**
>
> We greatly appreciate your valuable feedback. Thus, we have added generated results to the project page to showcase the effectiveness of DisPose.
>
> > **Q1: Real-life human results.**
>
> As discussed in Section 8 of the manuscript, we provide the qualitative results of the synthetic characters to avoid leakage of personal privacy. Thanks for your thoughtful comments. We provide real-life results on the project page (https://anonymous.4open.science/r/DisPose-AB1D). Please do not distribute or divulge these personal portraits.
>
> > **Q2: Optical flow may introduce error information from the noise background.**
>
> 1. First of all, we would like to clarify our setting. In this paper, we focus on controllable human animation. Following prior work (Animate Anyone[1]), given a static image and a driving video, we aim to generate the corresponding animation video. It belongs to the field of image-to-video generation. Specifically, it is required to preserve a static background, which is consistent with the reference image. Therefore this task does not support accepting dynamic backgrounds. On the other hand, there are some "similar" video generation works that allow taking noisy backgrounds, e.g., MIMO[2], and MotionEditor[3], which implement controllable character video generation as a video editing task, i.e., the input for video editing is a reference video and reference characters. They aim to replace the characters in the reference video using the reference characters.
>
> 2. Moreover, our method does not introduce background noise. Extracting the optical flow directly from the driving video may capture incorrect background noise. In contrast, our motion field estimation is based entirely on the skeleton pose through motion propagation, which makes the extracted motion field only relevant to the character's motion and avoids introducing noise in the background.
>
> [1] Animate Anyone: Consistent and Controllable Image-to-Video Synthesis for Character Animation. CVPR2024.
>
> [2] MIMO: Controllable Character Video Synthesis with Spatial Decomposed Modeling. arXiv2024.
>
> [3] MotionEditor: Editing Video Motion via Content-Aware Diffusion. CVPR2024.
>
> > **Q3: Evaluating models to preserve subject identity while transferring test poses.**
>
> We appreciate your thoughtful consideration. Considering the similarity of topics in the Tiktok dataset, we did not use the videos in the Tiktok dataset to train our model. To further compare the generalization ability of the models, we provide comparison experiments for cross-pose and cross-identity on the project page. Compared to the baseline model, our method improves overall quality, background consistency, and character consistency.

---

> > ### Comment · Reviewer_ZwXk · 2024-11-24
> >
> > Thanks for the detailed responses, I appreciate the efforts, and most of my concerns are addressed.
> >
> > I am not entirely convinced that the statement "our method does not introduce background noise" is true. In Figure 2., we can see part of the background included in the motion fields. If the method really won't include any background noises, then it should work on reference videos with noisy/dynamic backgrounds, since only the character motions would be preserved, as suggested in the rebuttal.
> >
> > That said, I agree that the proposed optical flow extraction should help address the noisy background issues to some extent, and, in my opinion, a method doesn’t need to be flawless to merit publication. Including this discussion in the manuscript should be sufficient.

---

> > > ### Author Response · Authors · 2024-11-25
> > >
> > > Thank you for your valuable comments. We have added Figure 15 in the Appendix H. Since our motion fields are not extracted directly from the driving video, some noise due to estimation errors may be introduced. The motion field of the reference image without the background is more accurate than the complex background.
> > >
> > > We would be eager to hear if you have any remaining feedback.

---

> > > > ### Comment · Reviewer_ZwXk · 2024-11-25
> > > >
> > > > Thanks for the additional analysis, I appreciate your efforts and prompt replies.
> > > > One last comment: line 862: "As shown in Figure 15The motion field of the" -> "As shown in Figure 15, the motion field ...".
> > > >
> > > > Overall, DisPose shows consistent improvements when combining with different base approaches, and it handles videos from different domains reasonably well. I raised my score to 6 --- marginally above the acceptance threshold.

---

> > > > > ### Author Response · Authors · 2024-11-26
> > > > >
> > > > > We're glad the additional experiments addressed your questions. Thank you again for reviewing our work and providing thoughtful feedback.

---

### Official Review · Reviewer_BkAE · 2024-11-04

**Soundness:** 2
**Presentation:** 2
**Contribution:** 2
**Rating:** 5
**Confidence:** 5

**Summary:**

This paper introduces a novel method for animating static human images into realistic videos using driving videos, without relying on additional dense inputs such as depth maps or DensePose. The authors present DisPose, which disentangles pose guidance into motion field estimation and keypoint correspondence. Specifically, they generate a dense motion field from a sparse motion field derived from skeleton poses and the reference image, providing region-level dense guidance while maintaining generalization. Simultaneously, they extract diffusion features corresponding to keypoints in the reference image and transfer these features to the target pose, enhancing appearance consistency and identity preservation. This approach is implemented as a plug-and-play hybrid ControlNet, allowing integration into existing human animation models without additional modifications to model parameters. Extensive qualitative and quantitative experiments demonstrate that DisPose outperforms current methods in both motion alignment and appearance consistency, effectively balancing generalizability and control effectiveness in pose-driven human image animation.

**Strengths:**

The strengths of this paper lays in the following several three folds:

1. The paper provides a clear and detailed explanation of the proposed method, including the construction of the tracking matrix from skeleton poses, Gaussian filtering, conditional motion diffusion, and the hybrid control network architecture. As well, the paper is well-organized, guiding the reader logically through the introduction, related work, methodology, experiments, and conclusions. This structure facilitates comprehension and highlights the contributions of the work.

2. The paper introduces DisPose, a novel framework that disentangles pose control into motion field guidance and keypoint correspondence. This innovative approach departs from traditional methods that rely heavily on dense control inputs, offering a new perspective on pose-guided human animation.

3. The DisPose module is designed as a plug-and-play component that can be seamlessly integrated into existing human image animation models. This versatility enhances the performance of baseline models without the need for extensive retraining.

**Weaknesses:**

The weakness of this paper lays in the following several three folds. My final rating is based on the discussion, and I would like to raise my score if some of the major concerns can be solved.

1. The experimental evaluation is primarily conducted on the TikTok dataset and an additional unseen dataset comprising 30 human videos. The reliance on these datasets raises concerns about the method's generalizability. e.g. the full body, side view.

2. The impact of the hybrid control network architecture, conditional motion diffusion process, and other auxiliary modules is not analyzed.
The figures shows promising visual results, such as the complete fingers and vivid facial expressions, so what proposed components lead to such enhanced visual details and visual improvements?

3. The evaluation employs VBench metrics and Fréchet Video Distance (FVD), which may not fully capture perceptual quality. The paper does not report other standard metrics such as Structural Similarity Index (SSIM) or Peak Signal-to-Noise Ratio (PSNR) for a more objective assessment.

**Questions:**

See the weakness section.

---

> ### Author Response · Authors · 2024-11-21
> **Official  Response to Reviewer BkAE**
>
> We greatly appreciate your valuable feedback. Therefore, we have incorporated several additional experiments into the paper to showcase the effectiveness of DisPose. These newly added experiments are highlighted in $\color{red}{red}$ within the revised version of the paper.
>
> > **Q1: More full-body and side-view generation results.**
>
> Thanks to your valuable suggestions, we provide the generated full-body and side-view videos on the project page (https://anonymous.4open.science/r/DisPose-AB1D), demonstrating our method's generalizability.
>
> > **Q2: The impact of hybrid ControlNet architecture.**
>
> We show the impact of hybrid ControlNet architecture in the following table. Specifically, we design two variant architectures, (1) Exp1: inserting the motion field into the denoising network instead of the hybrid ControlNet as shown in Figure 12(a) of Appendix E.1, and (2) Exp2: removing the hybrid ControlNet and inserting the motion field guidance and keypoint correspondence into the denoising network as shown in Figure 12(b) of the Appendix E.1. We have updated these results in Table 4 of the manuscript.
>
> |Model      |FID-FVD|FVD    |
> | :---       | :---:   | :---:   |
> |Exp1       |10.43  |514.83 |
> |Exp2       |10.94  |551.32 |
> |Full Model |10.24  |466.93 |
>
> Exp1 shows that the motion field needs to be jointly optimized with U-Net to provide the correct representation. Exp2 shows that complex motion information and appearance features cannot be modeled with only two shallow encoders.
>
> > **Q3: The impact of CMP.**
>
> We provide the ablation analysis of CMP in the following table, which shows that CMP can improve the consistency of the generated video. We have updated these results in Table 5 of the manuscript.
>
> | Model             |subject_consistency    |background_consistency |
> |:---                |:---:                    |:---:                    |
> |Full Model w/o CMP |93.94                  |   97.83               |
> |Full Model         |94.35                  |   98.75               |
>
> > **Q4: Reasons for enhanced visual details and visual improvements.**
>
> We provide additional ablation analyses in Appendix C. The region-level guidance provided by our motion field guidance facilitates the enhancement of consistency across body regions. The proposed keypoints correspondence improves generation quality by aligning DIFT features of the skeleton pose, e.g., facial consistency.
>
> > **Q5: More standard metrics.**
>
> To further evaluate the generated results, we provide performance comparisons for image-level metrics in the table below. Compared to the baseline model, our method achieves significant improvements in all metrics. We have updated these results in Table 6 of the manuscript.
>
> |Model  |SSIM   |PSNR    |LPIPS |   L1|
> |:---                |:---:                    |:---:                    | :---: | :---: |
> |MusePose[1]    |0.788| 19.14|  0.263|  2.46E-05|
> |MusePose[1]+Ours   |0.811  |19.36  |0.238  |2.26E-05|
> |MimicMotion[2] |0.749  |18.32  |0.272| 2.71E-05|
> |MimicMotion[2]+Ours    |0.781  |19.58  |0.242  |2.42E-05|
>
> [1]MusePose: a Pose-Driven Image-to-Video Framework for Virtual Human Generation. 2024.
>
> [2]MimicMotion: High-Quality Human Motion Video Generation with Confidence-aware Pose Guidance. 2024.

---

> > ### Author Response · Authors · 2024-11-26
> >
> > Dear reviewer:
> >
> > As the discussion period is nearing its end, we hope to know whether our response has addressed your concerns to merit an increase in the rating, or if there are any issues that you would like us to clarify.
> >
> > Thank you once again for your time and effort.
> >
> > Sincerely
> > Authors of Paper 6301

---

> > > ### Author Response · Authors · 2024-12-01
> > >
> > > Dear Reviewer BkAE,
> > >
> > > We hope this email finds you well. With the discussion period coming to a close, We wanted to follow up and see if there are any additional questions or concerns about our paper or the rebuttal we provided earlier that we could help clarify.
> > >
> > > Your detailed feedback has been Insightful, and we have put significant effort into addressing the points you raised. If there are any remaining aspects where further clarification might strengthen your understanding of our work, we would be willing to provide more information.
> > >
> > > Furthermore, if you feel our responses have addressed your concerns effectively, we would greatly appreciate it if you would consider revisiting your initial rating of our submission. Your expert evaluation plays a crucial role in shaping the final outcome, and we sincerely appreciate your time and efforts throughout this review process.
> > >
> > > Thank you again for your dedication to improving the quality of submissions. Please let us know if there is anything else we can assist with.
> > >
> > > Best regards,
> > >
> > > Authors of Submission 6301

---

> > > > ### Comment · Area_Chair_KQy4 · 2024-12-01
> > > >
> > > > Dear Reviewer BkAE,
> > > >
> > > > Do the authors' responses address your concerns? It would be helpful if you could provide feedback so that we can understand whether any issues still remain.
> > > >
> > > > Thank you.
> > > >
> > > > AC

---

### Author Response · Authors · 2024-11-21
**General Response to ACs**

We sincerely thank the reviewers for their detailed and valuable feedback. All reviewers (BkAE, ZwXk, PDfA, c9da) appreciate the effectiveness and potential of DisPose.
Based on these comments, we have added some noteworthy responses for the reviewers, including:

- **[Reviewers BkAE and PDfA]** We added Table 4, Table 5, Figures 11 and 12, where they show more comprehensive ablation analysis.

- **[Reviewers BkAE and c9da]** We added Table 6, and Table 7 and updated Table 1. They show performance comparisons in more evaluation metrics and computational efficiency.

- **[Reviewers ZwXk, PDfA and c9da]** We clarified this paper's setting and more technical details.

- **[Reviewers BkAE, ZwXk, PDfA, c9da]** We added more generated results on the project page, including real-life characters, side-by-side videos, cross-pose and cross-identity comparisons.

We sincerely hope this work will shed some light on the field of controllable animation. Once again, we thank all the reviewers for spending their valuable time to help improve our work.

---

### Meta-Review · Area_Chair_KQy4 · 2024-12-18

**Metareview:**

This paper introduces a new framework, DisPose, for human pose retargeting that offers high-quality motion control. The main idea of this method is to disentangle the sparse skeleton pose in human image animation into motion field guidance and keypoint correspondence, which is achieved by generating a dense motion field from a sparse motion field and the reference image. The advantage of this method is that it provides region-level dense guidance while maintaining the generalization of sparse pose control.

The main weaknesses mentioned in the reviews are the insufficient experiments in demonstrating generalizability and the issues with the evaluation metric, FVD. The authors provide more realistic generation results, such as full-body and side-view generation, as well as out-of-domain real human identities. The revision also includes evaluations using standard metrics and the debiased FVD.

After the rebuttal, the paper received mostly positive reviews, with final ratings of 5, 6, 6, and 8. The area chair concurs with the reviewers' suggestions and recommends accepting the paper.

**Additional Comments On Reviewer Discussion:**

The authors have addressed the reviewers' concerns by providing additional video visualizations of
* side-by-side videos,
* cross-pose and cross-identity comparisons,
* generation results of real-life characters.

The revision also includes additional experiments regarding
* effectiveness of multi-scale point correspondence,
* evaluation with content debiased FVD,
* trainable parameters and inference time,
* more ablation study and analysis.

---

### Decision · Program_Chairs · 2025-01-22

Accept (Poster)